# IL-1β Controls Proliferation, Apoptosis, and Necroptosis Through the PI3K/AKT/Src/NF-κB Pathway in Leukaemic Lymphoblasts

**DOI:** 10.3390/biomedicines14010041

**Published:** 2025-12-24

**Authors:** Zitlal-Lin Victoria-Avila, Elba Reyes-Maldonado, María Lilia Domínguez-López, Jorge Vela-Ojeda, Aranza Lozada-Ruiz, Omar Rafael Alemán, Ruth Angélica Lezama

**Affiliations:** 1Escuela Nacional de Ciencias Biológicas, Instituto Politécnico Nacional Prolongación de Carpio y Plan de Ayala, Col. Santo Tomas, Mexico City 11340, Mexico; huitzitzillin85@gmail.com (Z.-L.V.-A.); elbareyesm@gmail.com (E.R.-M.); ldmguez@yahoo.com.mx (M.L.D.-L.); shelovega2010@gmail.com (J.V.-O.); ara.lozruiz@gmail.com (A.L.-R.); orafaelam@gmail.com (O.R.A.); 2Departamento de Inmunología, Instituto de Investigaciones Biomédicas, Universidad Nacional Autónoma de México, Mexico City 04510, Mexico

**Keywords:** leukaemia, IL-1β, RS4:11 cells, inflammation, proliferation, apoptosis, necroptosis

## Abstract

**Background:** Chronic inflammation and the development of cancer are closely linked, with components that comprise the tumour microenvironment—including proinflammatory cytokines—exerting essential tumourigenic effects. These proinflammatory cytokines include IL-1β, which has been reported to be overexpressed in several cancers and shown to activate several signalling pathways. These pathways may involve kinases such as AKT (serine/threonine kinase) and Src (Proto-oncogene tyrosine-protein kinase), and have a broad capacity to activate nuclear factors, including NF-κB (Nuclear Factor kappa-light-chain-enhancer of activated B cells), which can regulate the transcription of genes encoding proteins such as cIAP1 (Cellular Inhibitor of Apoptosis Protein 1), Bcl-2 (B-cell lymphoma 2), and cyclin D1, thereby regulating processes like apoptosis and cell cycle inhibition. **Objectives:** The aim of this study was to investigate the role of IL-1β (*Interleukin-1 beta*) in regulating cell death and proliferation in RS4:11 leukaemic lymphoblasts via the PI3K (Phosphoinositide 3-kinase)/AKT/Src/NF-κB pathway using an in vitro experimental approach. **Methods:** We employed flow cytometry to determine the expression levels and phosphorylation status of various proteins; proliferation was assessed using the CCK-8 kit, and apoptosis was evaluated with the Annexin V kit. **Results:** Our findings indicate that the IL-1β-activated signalling pathway modulates these cellular processes in leukaemic lymphoblasts. **Conclusions:** We therefore conclude that IL-1β exerts significant effects on cell death and proliferation in leukaemic lymphoblasts through the PI3K/AKT/NF-κB pathway, with the study’s findings indicating that an inflammatory environment may promote such lymphoblasts to acquire neoplastic characteristics. As such, the proteins involved in the effects evaluated in this work could be considered as potential therapeutic targets for the treatment of Acute Lymphoblastic Leukaemia (ALL).

## 1. Introduction

Inter- and intracellular events such as inhibition of apoptosis, proliferation, differentiation, and inflammation are involved in the development of neoplastic processes [1]. In the specific case of inflammation, the presence of proinflammatory cytokines such as IL-1β, IL-6 (Interleukin-6), and TNF-α (Tumour Necrosis Factor) has been reported in the tumour microenvironment of various cancers [2]. These cytokines are produced by immune cells that are attracted to the niche in which cancer cells develop and may also be secreted by the same cells that comprise the niche [3,4]. Increased IL-1β expression has been reported in several cancers, including hepatocellular carcinoma, colorectal cancer, and chronic myeloid leukaemia [5,6,7]. Once IL-1β is recognised by its receptor on the cell surface, it initiates a signalling cascade involving kinases and transcription factors, such as NF-κB, that regulate the expression of genes involved in apoptosis, proliferation, and inflammation [8].

ALL is a haematopoietic neoplasm that originates in the bone marrow and subsequently spreads to the lymph nodes, spleen, and central nervous system [9].

Although the effect of IL-1β on lymphoblasts from ALL has not been reported, it has been shown to induce NF-κB and PI3K activation in other cancers, such as liver cancer, and to influence NF-κB activation [10]. Another study indicated that Src kinase induces NF-κB activation and IL-6 expression in cells undergoing Src oncogene-dependent transformation [11]. The potential effects of NF-κB activation mentioned above may help neoplastic cells to survive and grow in the niche where they originated.

This study aimed to determine whether the inflammatory cytokine IL-1β, via kinases such as PI3K/AKT, Src, and NF-κB, influences cell death and proliferation in leukaemic lymphoblasts. We observed that this cytokine inhibits apoptosis and limits necroptosis while promoting proliferation in leukaemic lymphoblasts, mediated by the activity and activation of PI3K, AKT, Src, and NF-κB.

## 2. Materials and Methods

In this study, the RS4:11 cell line—obtained from the ATCC, a recognised biological resource centre—was used as the cellular model. RS4:11 cells were originally derived from the bone marrow of a 32-year-old female patient with relapsed ALL. These cells exhibit a biphenotypic profile, co-expressing both B-cell and monocytic markers, indicating mixed-lineage characteristics typical of this leukaemia subtype. Additionally, RS4:11 cells display features consistent with pre-B lymphoblasts, including expression of CD19 (Cluster of Differentiation 19), HLA-DR (Human Leukocyte Antigen–DR isotype), and terminal deoxynucleotidyl transferase. These cells also demonstrate lineage plasticity upon differentiation induction, making RS4:11 a robust and valuable model for investigating signalling pathways and evaluating therapeutic strategies in acute lymphoblastic leukaemia [12,13].

### 2.1. Cell Culture

The RS4:11 cell line was cultured in Iscove’s medium supplemented with 10% FBS and 1% penicillin–streptomycin at 37 °C with 5% CO_2_. Lymphoblasts were harvested, washed with PBS (*Phosphate-Buffered Saline*), and then seeded overnight in 48-well plates with serum-free culture medium at a cell density of 1.0 × 10^6^ lymphoblasts/mL. The cells were treated with 10 nM of IL-1β (Peprotech, Cranbury, NJ, USA), and the following inhibitors were used: for PI3K, Wortmannin at 200 nM [14]; for Src, PP2 at 20 μM [15] (Sigma Co, St. Louis, MO, USA); and for NF-κB, JSH23 (Calbiochem, San Diego, CA, USA) at 10 µM [16].

All inhibitors were used at non-toxic concentrations, as reported in the references cited above. To verify lymphoblast viability at all culture time points in the presence of each inhibitor, trypan blue exclusion assays were performed, demonstrating 95–99% cell viability.

### 2.2. Protein EXPRESSION via Flow Cytometry

The RS4:11 cells were washed with PBS and then fixed with 2% paraformaldehyde at 37 °C for 10 min. They were then permeabilised with 1% SDS (Sodium Dodecyl Sulfate) for 5 min, followed by additional washing with PBS. Next, the cells were incubated overnight at 4 °C with the corresponding primary antibodies (Santa Cruz Biotechnology, Santa Cruz, CA, USA) at a dilution of 1:150 each for anti-p-AKT (SC-293125), anti-p-Src (SC-101802), P-p65 (SC-136548), Bcl-2 (SC-7382), cIAP1 (SC-271419), and cyclin D1 antibody (SC-20044), at a final concentration of 1.3 µg/µL. After removing the primary antibodies by washing with PBS, the cells were incubated for 1 h at room temperature with a FITC (Fluorescein Isothiocyanate) labelled secondary antibody (West Grove, PA, USA, 115-095-166, 111-095-144) diluted 1:100, followed by additional washing with PBS. Non-specific binding was assessed by incubating the cells with the secondary antibody alone. Detection of FITC-positive leukaemic lymphoblasts was performed using an Aria FACS cytometer (Becton Dickinson, Franklin Lakes, NJ, USA) analysing 10,000 events, and the data were analysed using the FlowJo software, version 10.0.

### 2.3. Proliferation and Apoptosis

Leukaemic lymphoblasts were cultured in 96-well plates, and cell proliferation was measured using the CCK-8 Kit (Enzo, Farmingdale, NY, USA). A suspension containing 2500 viable cells per well was seeded into 96-well plates and incubated with IL-1β and the respective inhibitors for 24, 48, or 72 h in a humidified incubator (37 °C, 5% CO_2_). Subsequently, 10 μL of the CCK-8 solution was added to each well, and the plates were incubated for an additional 2 h. Absorbance was then measured at 450 nm using a microplate reader.

For apoptosis analysis, lymphoblasts were incubated with IL-1β for 48 h, then washed and stained with Annexin V–FITC/PI (*Propidium Iodide*). After treatment with IL-1β and the respective inhibitors, 1–5 × 10^5^ cells were collected by centrifugation and resuspended in 500 μL of Binding Buffer II. Next, 5 μL of Annexin V–FITC II and 5 μL of PI II were added, and the samples were incubated for 5 min at room temperature in the dark. Each sample was analysed on a flow cytometer (FACS Aria) Becton, Dickinson Co., Franklin Lakes, NJ, USA, and data were processed using FlowJo software to determine the proportions of necrotic cells (Annexin V–FITC^−^/PI^+^), early apoptotic cells (Annexin V–FITC^+^/PI^−^), and late apoptotic cells (Annexin V–FITC^+^/PI^+^).

### 2.4. Statistical Analysis

Three independent experiments were performed in triplicate for each measurement. Data are shown as the mean ± standard deviation. Statistical analyses were carried out using GraphPad Prism version 5.0. One-way analysis of variance (ANOVA) was used to evaluate the phosphorylation levels of AKT, Src, and p65 (NF-κB p65 subunit) in cells treated with IL-1β alone or combined with inhibitors. Tukey’s multiple comparison test was applied to compare the expression of Bcl-2, cIAP1, and Cyclin D1 under these conditions, aiming to determine statistically significant differences between groups. Additionally, Student’s *t*-test was used to analyse apoptosis and proliferation.

Continuous variables were compared using parametric tests after confirming normality with the Shapiro–Wilk test (*p* > 0.05 in all cases). A *p*-value < 0.05 was considered statistically significant.

## 3. Results

### 3.1. IL-1β Induces AKT, Src, and NF-κB Activation in RS4:11 Leukaemic Lymphoblasts

We examined whether PI3K/AKT and Src were activated by IL-1β at different culture times (24, 48, and 72 h) by assessing the phosphorylation of serine 473 on AKT and tyrosine 416 on Src through flow cytometry. When measuring AKT phosphorylation, we found a 4-fold increase compared to the control group after 24 h (9306 vs. 2296 MFI); at 48 and 72 h, the increase was 2.9-fold (4268 vs. 1454 MFI and 2860 vs. 956 MFI, respectively; see Figure 1A,B).

To test whether IL-1β activates PI3K/AKT, and since PI3K activates AKT, we used wortmannin to inhibit PI3K (and, consequently, AKT) in leukaemic lymphoblasts cultured with IL-1β for 24 h. We found that AKT phosphorylation decreased 2.53-fold in the presence of wortmannin compared with lymphoblasts treated with IL-1β alone (3945 vs. 9306 MFI; Figure 1C,D).

Regarding Src activation induced by IL-1β in leukaemic lymphoblasts, we observed a 2-fold increase at 24 h (2006 vs. 965 MFI) and a 3 and 4-fold increase at 48 and 72 h compared to the control group (1290 vs. 430 MFI and 1500 vs. 350 MFI, respectively; Figure 1E,F. To determine whether IL-1β triggers Src activation—observed as the highest MFI of Src phosphorylation after 24 h of culture—leukaemic lymphoblasts were incubated with IL-1β at the same time point in the presence of a Src inhibitor. Src phosphorylation was significantly reduced when cells were treated with PP2, compared to those treated with IL-1β alone (2006 vs. 1100 MFI, 0.55-fold decrease; Figure 1G,H). 

To assess NF-κB activation induced by IL-1β, leukaemic lymphoblasts were cultured for 24, 48, and 72 h in the presence of IL-1β, and phosphorylation of serine 536 on the p65 subunit of NF-κB was measured using flow cytometry. We observed that after 24 h of incubation, p65 phosphorylation increased 3.7-fold (12,384 vs. 3281 MFI); after 48 h, it was 1.66-fold higher (5507 vs. 2166 MFI); and, after 72 h, it was 2.9-fold higher (4953 vs. 1107 MFI); see Figure 1J.

To examine whether IL-1β-induced activation of NF-κB involves PI3K and Src kinases, lymphoblasts were cultured for 24 h with IL-1β and inhibitors targeting these proteins. We observed that p65 phosphorylation was significantly reduced by both inhibitors compared to IL-1β alone (8391 MFI vs. 12,384 MFI, 1.47-fold decrease and 9544 vs. 12,384 MFI, 1.29-fold decrease, respectively; Figure 1K,L). To confirm that IL-1β specifically triggers NF-κB activation, lymphoblasts were treated with the NF-κB inhibitor for 1 h before and with IL-1β for 24 h after. A significant decrease in MFI corresponding to p65 phosphorylation was noted (6609 vs. 12,384 MFI, 1.87-fold decrease; Figure 1K,L). 

### 3.2. IL-1β Influences Expression of Bcl-2 and cIAP1 via PI3K/AKT/NF-κB Pathway in RS4:11 Leukaemic Lymphoblasts

The genes encoding the anti-apoptotic proteins Bcl-2 and cIAP are known to be regulated by NF-κB [17,18]. Therefore, considering our previous results, we examined whether activation of the PI3K/AKT/NF-κB pathway due to IL-1β affected Bcl-2 and cIAP expression in leukaemic lymphoblasts. To do this, we cultured leukaemic lymphoblasts with IL-1β for 24, 48, and 72 h and measured Bcl-2 and cIAP levels using flow cytometry.

When measuring the effect of IL-1β on Bcl-2 expression, we did not see a significant increase compared with control lymphoblasts at 24 h (2367 vs. 2047 MFI). However, a 2.2-fold increase was observed at 48 h (4754 vs. 2141 MFI), and a 1.9-fold increase at 72 h (3419 vs. 1775 MFI); see Figure 2A,B. Additionally, to determine whether IL-1β induces Bcl-2 expression Via PI3K/AKT, Src, and NF-κB, we cultured leukaemic lymphoblasts with IL-1β in the presence of inhibitors targeting these pathways. Since the most significant increase in Bcl-2 expression was seen at 48 h, we cultured the lymphoblasts for the same period. Inhibition of PI3K, Src, and NF-κB reduced Bcl-2 expression by 3.9-fold (1995 vs. 4754 MFI), 2-fold (2361 vs. 4754 MFI), and 2.9-fold (1610 vs. 4754 MFI), respectively; see Figure 2C,D.

Regarding cIAP1 expression, we observed that IL-1β in leukaemic lymphoblasts induced a 3.7-fold (13,661 vs. 3601 MFI), a 3.3-fold (7128 vs. 2144 MFI), and a 3.8-fold (8717 vs. 2291 MFI) increase at 24, 48, and 72 h of culture, respectively; see Figure 2E,F. 

To determine whether PI3K, Src, and NF-κB regulate cIAP1 expression, we also cultured leukaemic lymphoblasts with IL-1β and inhibitors of these proteins. Since cIAP1 expression was similar across all three culture durations, we chose to culture the lymphoblasts for 24 h. Wortmannin, PP2, and JSH23 decreased cIAP1 expression by 2.2-fold (5954 vs. 13,661 MFI), 1.79-fold (7631 vs. 13,661 MFI), and 3.5-fold (3824 vs. 13,661 MFI), respectively; see Figure 2G,H. 

### 3.3. IL-1β Decreases Apoptosis and Limits Necroptosis via the PI3K/AKT/Src/NF-κB Pathway in RS4:11 Leukaemic Lymphoblasts

Having observed that IL-1β increases the expression of anti-apoptotic proteins in leukaemic lymphoblasts, we investigated whether this interleukin inhibits apoptosis in these cells. To do this, we analysed the presence of phosphatidylserine residues on lymphoblast cell surfaces by labelling with annexin V after 48 h, as we observed the highest expression of Bcl-2 at this time. 

As shown in Figure 3A and Table 1 and Appendix A, lymphoblasts treated with IL-1β had a 37% higher viable cell rate than control lymphoblasts (89.76% vs. 65.11%). We also observed that these lymphoblasts exhibited a 5.3-fold decrease in early apoptosis compared with untreated lymphoblasts (0.90% vs. 5.0%). Regarding lymphoblasts in late apoptosis, we observed a similar percentage in both experimental groups, although they exhibit a significant difference (IL-1β: 0.91%; control group: 0.84%; see Figure 3A, Table 1 and Appendix A).

In our analysis of the involvement of the transcription factor NF-κB and the kinases Src and AKT in early apoptosis in leukaemic lymphoblasts, we found that it decreased by 1.7-fold when NF-κB was inhibited (0.90% vs. 0.53%), by 3.9-fold when PI3K was inhibited (0.90% vs. 0.24%), and by 1.5-fold when Src was inhibited (0.90% vs. 0.62%); see Figure 3B and Table 1 and Appendix A. Inhibiting these same proteins in the presence of IL-1β had a less pronounced effect on late apoptosis in leukaemic lymphoblasts: wortmannin decreased it by 1.2-fold (from 0.91% to 0.75%), while PP2 decreased it by 1.0-fold (from 0.91% to 0.83%). However, when NF-κB was inhibited in leukaemic lymphoblasts in the presence of IL-1β, we observed a 2-fold increase in late apoptosis compared to lymphoblasts treated only with this interleukin (1.9% vs. 0.91%); see Figure 3B and Table 1 and Appendix A.

Regarding necrotic cells, we observed that lymphoblasts treated with IL-1β exhibited 5.0% necrotic cell death, compared to 13.64% in the control group (Figure 3A, Table 1), indicating that IL-1β reduces necrosis by 36.9%. Concerning treatment with inhibitors, we found that when lymphoblasts were treated with IL-1β plus JSH3, wortmannin, or PP2, the percentage of necrotic lymphoblasts was 30.5%, 12.7%, and 10.53%, respectively. These results show that inhibiting NF-κB, PI3K, and Src in these lymphoblasts increases necrosis by 6-fold, 2.5-fold, and 2.1-fold, respectively; see Table 1 and Appendix A. 

### 3.4. IL-1β via PI3K/AKT/Src/NF-κB Influences Cyclin D1 Expression and Proliferation of RS4:11 Leukaemic Lymphoblasts

Cyclin D1 overexpression has been observed in various types of cancer [19]; however, it remains unclear whether IL-1β can influence cyclin D1 expression. Therefore, our study assessed the potential effect of IL-1β on cyclin D1 expression in leukaemic lymphoblasts through the PI3K/AKT/Src/NF-κB pathway at 24, 48, and 72 h of culture. When measuring cyclin D1 protein levels after IL-1β stimulation in leukaemic lymphoblasts, we observed a 1.5-fold increase at 24 h compared to the control group (6420 vs. 4192 MFI), a 1.85-fold increase at 48 h (7076 vs. 3813 MFI), and a 2.3-fold increase at 72 h (12,905 vs. 5381 MFI); see Figure 4A,B.

Subsequently, since we observed the highest upregulation of cyclin D1 in leukaemic lymphoblasts at 72 h of culture with IL-1β, we investigated whether activation of AKT, Src, and NF-κB could influence cyclin D1 expression after this period. Inhibition of PI3K, Src, and NF-κB resulted in 1.98-fold (6493 vs. 12,905 MFI), 1.78-fold (7247 vs. 12,905 MFI), and 2.98-fold (4328 vs. 12,905 MFI) decreases in cyclin D1 expression, respectively, as shown in Figure 4C,D. 

Since we observed that IL-1β induces overexpression of cyclin D1 in leukaemic lymphoblasts Via the PI3K/AKT/Src/NF-κB pathway, we first examined whether IL-1β promotes proliferation of leukaemic lymphoblasts at 24, 48, and 72 h of culture. As shown in Figure 4E–G, IL-1β levels increase by 81%, 110%, and 96% compared to the control at 24, 48, and 72 h, respectively. We then evaluated the effects of IL-1β on the proliferation of leukaemic lymphoblasts at these time points, using specific inhibitors for NF-κB, PI3K, and Src. At all time points, proliferation was reduced compared to lymphoblasts treated only with IL-1β (at 24 h: NF-κB, 87%; PI3K, 71%; Src, 58%). At 48 h: NF-κB, 91%; PI3K, 90%; Src, 43%. At 72 h: NF-κB, 93%; PI3K, 92%; Src, 46%; see Figure 4E–G).

## 4. Discussion

As they secrete molecules like proinflammatory cytokines and interferons, promoting a chronic inflammatory state that supports the formation of a tumour microenvironment, the cells that make up the immune system play a vital role in the development and progression of cancer [20]. Increased IL-1β expression has been seen in chronic myeloid leukaemia [21], and autocrine secretion of IL-1β has been reported in cell lines derived from chronic monocytic leukaemia and ALL [22]. Reports have shown that IL-1β is linked to PI3K/AKT activation in various cancers [23]. Additionally, in 2007, Lin highlighted the connection between IL-6 and Src in gastric cancer [24]. Based on this evidence, we examined whether IL-1β activated AKT and Src in lymphoblasts derived from ALL and found that this interleukin significantly increased the activation of both kinases. Additionally, this result indirectly indicates that these lymphoblasts express the IL-1β receptor. Therefore, we examined whether exposure to IL-1β could trigger NF-κB activation and found that this interleukin elicited the most significant increase in NF-κB activity in RS4:11 leukaemic lymphoblasts. This aligns with findings in breast cancer, where IL-1β and NF-κB are linked to the disease’s progression [25].

These results led us to consider a potential signalling pathway in which IL-1β could activate NF-κB. Then, we tested whether the activation of these kinases, caused by IL-1β, might be involved in NF-κB activation in lymphoblasts. We found that inhibiting PI3K and Src significantly reduced NF-κB activation. It has been reported separately in breast and cervical cancer cell lines that PI3K–AKT activates NF-κB, and in liver cancer that Src is involved in activating this nuclear factor [26,27,28]; therefore, these reports support our findings. However, in this study, we demonstrated in the same ALL cell model that IL-1β activates a signalling pathway involving PI3K/AKT/Src/NF-κB—an observation that has not yet been reported for any leukaemia.

On the other hand, it has previously been suggested that a general inflammatory environment may be linked to some inherent characteristics of neoplastic processes, such as the suppression of apoptosis and increased proliferation [29]. Therefore, we examined whether IL-1β alters the expression of anti-apoptotic proteins such as Bcl-2 and cIAP1 in leukaemic lymphoblasts. We found that IL-1β significantly increases the expression of Bcl-2 and cIAP1 in these lymphoblasts; additionally, when PI3K, Src, and NF-κB were inhibited, the levels of these anti-apoptotic proteins decreased markedly.

Similarly, we show that the most significant increase in Bcl-2 expression occurred at 48 h in the presence of IL-1β, while the highest activation of AKT, Src, and NF-κB was observed at 24 h. Although activation of these proteins remained slightly elevated at 48 and 72 h (with the increase being statistically significant), this pattern suggests that Bcl-2 upregulation may be associated with sustained activation of AKT, Src, and NF-κB between 24 and 72 h.

For cIAP1, peak expression occurred at 24 h, coinciding with the maximum activation of AKT, Src, and NF-κB. Although cIAP1 levels remained elevated at 48 and 72 h, these changes were also statistically significant. These findings suggest that IL-1β-induced cIAP1 expression initially corresponds to increased activation of AKT, Src, and NF-κB, followed by sustained activity of Src and NF-κB. Overall, these observations indicate that IL-1β regulates Bcl-2 and cIAP1 expression through the AKT–Src/NF-κB signalling pathway in leukaemic lymphoblasts.

Several research groups have reported the presence of these anti-apoptotic proteins in various types of neoplasms [30,31,32], suggesting that they may regulate apoptosis. Regarding the relationship between proinflammatory interleukins and apoptosis-regulating proteins, it has been observed that IL-1β induces phosphorylation of Bcl-2 in acute myeloid leukaemia blasts [21]; on the other hand, in 2002, Spets et al. found that IL-6 (Interleukin-6) induces an increase in the expression of anti-apoptotic proteins such as Bcl-2 and MCL-1 (Myeloid Cell Leukaemia 1) in multiple myeloma cells [33]. These data support our finding that IL-1β is involved in the expression of anti-apoptotic proteins in leukaemic lymphoblasts.

Considering these results, we examined whether this interleukin affects the death of leukaemic lymphoblasts. Our findings showed that IL-1β significantly reduces the number of lymphoblasts undergoing early apoptosis. These results align with previous research, which reports that Bcl-2 inhibits early apoptosis by preventing the externalisation of phosphatidylserine on the plasma membrane and by blocking cytochrome c release, thereby inhibiting caspase activity. It is well known that cIAP1 inhibits caspases, which function in the initial stage of apoptosis [34,35]. This observation is consistent with the findings of Turzanki et al. in 2004, who reported that IL-1β produced by blast cells from patients with chronic myeloid leukaemia induces resistance to apoptosis [21].

Regarding late apoptosis, no significant differences were observed between leukaemic lymphoblasts treated with or without IL-1β, nor were any differences found between early and late apoptosis in the presence of IL-1β. Taken together, these findings suggest that IL-1β has a general inhibitory effect on apoptosis in leukaemic lymphoblasts.

Conversely, inhibiting NF-κB, PI3K, and Src in leukaemic lymphoblasts with IL-1β present resulted in a general decrease in the percentage of lymphoblasts undergoing early apoptosis. These findings support our previous observations about how these kinases and transcription factors affect the expression of Bcl-2 and cIAP1. 

Regarding the analysis of late apoptosis in leukaemic lymphoblasts treated with IL-1β in the presence of PI3K and Src inhibitors, we observe no significant difference in lymphoblasts at this stage of apoptosis—likely because IL-1β does not promote late apoptosis. Conversely, we found that leukaemic lymphoblasts treated with IL-1β alongside the NF-κB inhibitor showed a modest increase in the percentage of lymphoblasts undergoing late apoptosis; since this factor is known to regulate the transcription of proteins involved in initiating apoptosis [8], this increase could be because, when this transcription factor is inhibited, apoptosis is no longer suppressed, allowing lymphoblasts to proceed with it. Therefore, these data also support our observation that IL-1β promotes early apoptosis in leukaemic lymphoblasts Via NF-κB.

Likewise, we observed that this interleukin restricts necrosis in leukaemic lymphoblasts. As is widely recognised, the concept of necrosis has developed into a highly regulated process, now known as necroptosis. Necroptosis involves kinases such as RIPK1, 2, and 3 (Receptor-Interacting Protein Kinase), as well as MLKL (Mixed Lineage Kinase Domain-Like protein) [36]. Since cIAP1 has a ubiquitin ligase domain, it can promote the degradation of target proteins [37]; one of these is RIPK1, which—when ubiquitinated and broken down by this pathway—can lead to a reduction in necroptosis [38]. In this study, we observed that IL-1β increased cIAP1 expression in leukaemic lymphoblasts. On one hand, this protein might prevent early apoptosis; on the other hand, it could limit necrosis by removing RIPK1. It is important to note that RIPK3 and MLKL have been reported in ALL cells, and when they were genetically suppressed, and caspase inhibitors were supplemented, a significant reduction in cell death was observed [39]. Feldmann et al. (2017) reported that in ALL cells, sorafenib—a multi-targeted tyrosine kinase inhibitor used for treating acute leukaemia—decreased MLKL phosphorylation, leading to reduced cell death [40]. These findings support our hypothesis and suggest that analysing RIPK1 and RIPK3 expression in our cell model at a later stage could provide further insights.

When we observed the necroptosis of lymphoblasts previously treated with PI3K, Src, and NF-κB inhibitors followed by IL-1β, we found that necroptosis increased under all these conditions compared to lymphoblasts treated only with IL-1β. However, the most significant increase was noted when NF-κB was directly inhibited. These results may be due to NF-κB targeting cIAP1 as one of its genes; therefore, inhibiting this transcription factor also reduces the expression of its target genes, including cIAP1 (observed in this study), which (as mentioned above) probably affects the elimination of RIPK1, leading to an increase in necroptosis. This effect was similarly observed when PI3K and Src kinases were inhibited, as these kinases influence the activation of this transcription factor. These results further suggest that the necrosis seen when lymphoblasts were treated with IL-1β was affected by activation of the PI3K/AKT/NF-κB signalling pathway, which indicates regulation of this pathway. This finding allows us to consider the observed necrosis as necroptosis, as the defining feature that differentiates necroptosis from necrosis is that the former is a regulated process. In this regard, we emphasise that our work demonstrates that IL-1β prevents cell death by inhibiting early apoptosis and limiting necroptosis in leukaemic lymphoblasts—a phenomenon that, to the best of our knowledge, has not been previously reported.

In this study, we also investigated whether IL-1β could affect the expression of proteins involved in cell cycle regulation, such as cyclin D1. It has not previously been established that IL-1β causes an increase in cyclin D expression in any disease or pathology; however, Bousserouel et al. (2004) reported that IL-1β slightly increased cyclin D in an in vitro system using rat muscle cells, and this effect was significantly enhanced with the addition of arachidonic acid [41]. It has also been reported that IL-7 (Interleukin-7) promotes cyclin D1 overexpression in lung cancer-derived cell lines [42]. These data support our finding that a proinflammatory interleukin—specifically, IL-1β—induces increased cyclin D1 expression in leukaemic lymphoblasts. On the other hand, PI3K inhibition in myeloma and breast cancer cells decreased cyclin D1 expression [19,43]. These data support our observation that PI3K activation, triggered by IL-1β in leukaemic lymphoblasts, plays a significant role in cyclin D1 expression.

Different researchers have found that NF-κB induces the transcription of the cyclin D1 gene [44], supporting our observation that inhibiting NF-κB decreases cyclin D1 expression in IL-1β-treated lymphoblasts. Furthermore, it is known that NF-κB activity can be regulated by PI3K and Src kinases [23], aligning with our finding that lymphoblasts exposed to IL-1β and inhibitors of these kinases show a significant reduction in cyclin D1 expression. Furthermore, the observation that cyclin D expression peaked at 72 h of IL-1β treatment in leukaemic lymphoblasts may be linked to the sustained activation of PI3K and Src, which can promote NF-κB activation and, consequently, induce cyclin D1 gene expression. This suggests that IL-1β may drive cell-cycle progression in lymphoblasts during this time frame.

We then tested whether this interleukin affected the proliferation of these lymphoblasts and found that it significantly increased lymphoblast growth. Evidence from both in vivo and in vitro models has shown that SPP1^+^ (*Secreted Phosphoprotein 1–positive*) macrophages release IL-1β, which then promotes the proliferation of head and neck squamous cell carcinoma [45]. Similarly, Teixeira et al. (2020) observed that silencing the NF-κB1 gene reduced proliferation by lowering IL-1β levels in renal cell carcinoma [46]. These findings support our conclusion that IL-1β can trigger lymphoblast proliferation; in our model, this effect is likely linked to increased Cyclin D1 expression.

Finally, we found that inhibiting PI3K, Src, and NF-κB reduces lymphoblast proliferation in response to IL-1β. These findings can also be explained by the fact that these kinases are involved in NF-κB activation, which, in turn, induces cyclin D1 expression. However, it is important to note that the highest inhibition of cyclin expression in leukaemic lymphoblasts treated with IL-1β was observed when both NF-κB and PI3K were inhibited. This aligns with reports in other cancer types regarding the regulation of Cyclin D by PI3K and NF-κB [43,44]. Regarding Src kinase inhibition, blocking this kinase in leukaemic lymphoblasts treated with IL-1β reduced cyclin expression by an average of 50%, indicating that IL-1β does not fully activate this kinase.

Considering all our results and the data reported in the literature, we propose a signalling pathway in leukaemic lymphoblasts triggered by IL-1β, involving PI3K/AKT/Src/NF-κB, that regulates cell death and lymphoblast proliferation. This pathway can be summarised as follows: in the presence of IL-1β, the PI3K/AKT and Src kinases are activated in leukaemic lymphoblasts, leading to the activation of IKKβ. The activation of IKKβ (*Inhibitor of Nuclear Factor κB Kinase subunit beta*) causes p65 translocation to the nucleus [47] and promotes the transcription of its target genes, including Cyclin D1, Bcl-2, and cIAP1 [48]. The increase in Cyclin D1 expression could contribute to the observed increase in proliferation. Similarly, the elevated levels of Bcl-2 and cIAP1 might be related to the reduction in early apoptosis, and cIAP1 may also help limit necroptosis through ubiquitination and subsequent degradation of RIPK1 (see Figure 5).

The presence of an inflammatory environment in the cellular niche where leukaemia cells appear or develop may contribute to their neoplastic characteristics. The role of proinflammatory cytokines and the proteins involved in the signalling pathways triggered by these molecules are related to the malignant abilities of these cells; therefore, this field of research points to potential therapeutic targets that could be very beneficial in treating patients with this or other types of neoplasm.

Finally, our work shows how a proinflammatory cytokine can reduce apoptosis and necroptosis while also increasing the proliferation of leukaemic lymphoblasts through the PI3K/AKT/NF-κB pathway. However, it is important to determine whether other proinflammatory cytokines, such as IL-6 or TNF-α, have similar effects and through which signalling pathways they may operate. Additionally, understanding the broader effects of proinflammatory cytokines is essential; for example, whether they play a role in epithelial–mesenchymal transition, which could allow leukaemic lymphoblasts to migrate to other organs and promote metastasis. Nevertheless, this remains a hypothesis that needs validation.

## Figures and Tables

**Figure 1 biomedicines-14-00041-f001:**
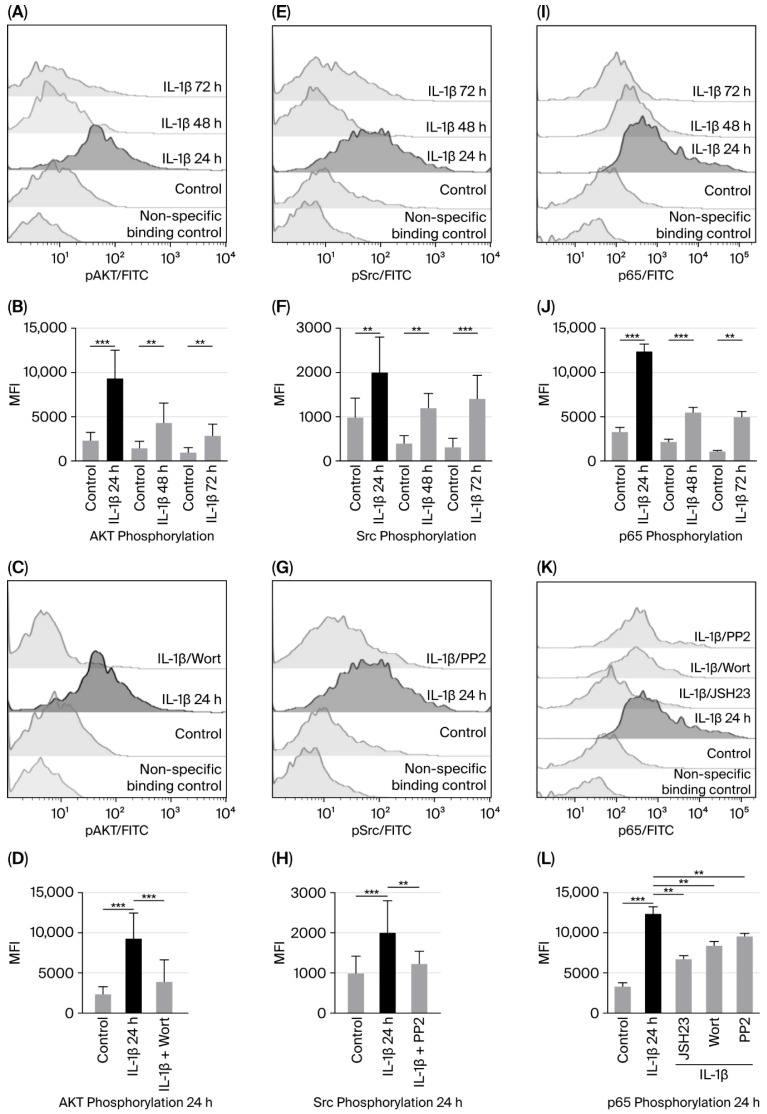
IL-1β induces increased activation of AKT, Src, and NF-κB in RS4:11 leukaemic lymphoblasts. (**A**,**E**,**I**) Representative flow cytometry histograms showing phosphorylation of AKT, Src, and the p65 subunit of NF-κB in leukaemic lymphoblasts at 24, 48, and 72 h of culture with IL-1β. (**B**,**F**,**J**) MFI analysis of phosphorylated AKT, Src, and p65 at these time points. (**C**,**G**,**K**) Representative histograms illustrating phosphorylation of AKT, Src, and p65 after 24 h of culture with IL-1β in the presence of NF-κB inhibitor (JSH23), PI3K inhibitor (wortmannin) and Src inhibitor (PP2), and (**D**,**H**,**L**) MFI analysis of AKT, Src, and phosphorylated p65 after 24 h of culture with IL-1β in the presence of these inhibitors. All results are expressed as the mean from three independent experiments, each performed in triplicate. *** *p* < 0.01 vs. control, ** *p* < 0.02 vs. control.

**Figure 2 biomedicines-14-00041-f002:**
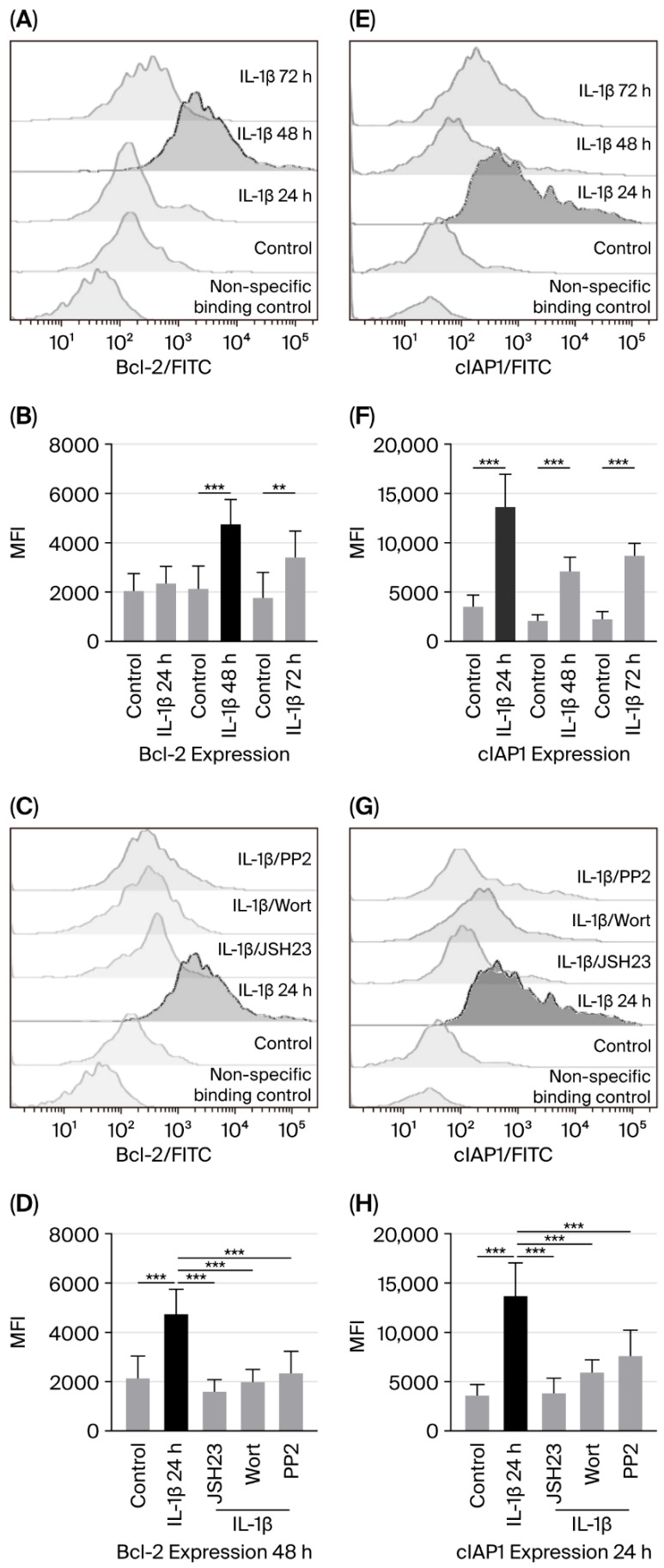
IL-1β causes overexpression of Bcl-2 and cIAP1 in RS4:11 leukaemic lymphoblasts through the PI3K/AKT/NF-κB pathway. (**A**,**E**) Representative flow cytometry histograms showing Bcl-2 and cIAP1 levels at 24, 48, and 72 h after IL-1β treatment. (**B**,**F**) MFI analysis of Bcl-2 and cIAP1 at those same time points with IL-1β. (**C**,**G**) Histogram of Bcl-2 expression at 48 h with IL-1β, in the presence of inhibitors JSH23, wortmannin and PP2. (**D**,**H**) MFI analysis of Bcl-2 at 48 h and cIAP1 at 24 h under the same conditions. All data represents three independent experiments performed in triplicate. *** *p* < 0.01 vs. control, ** *p* < 0.02 vs. control.

**Figure 3 biomedicines-14-00041-f003:**
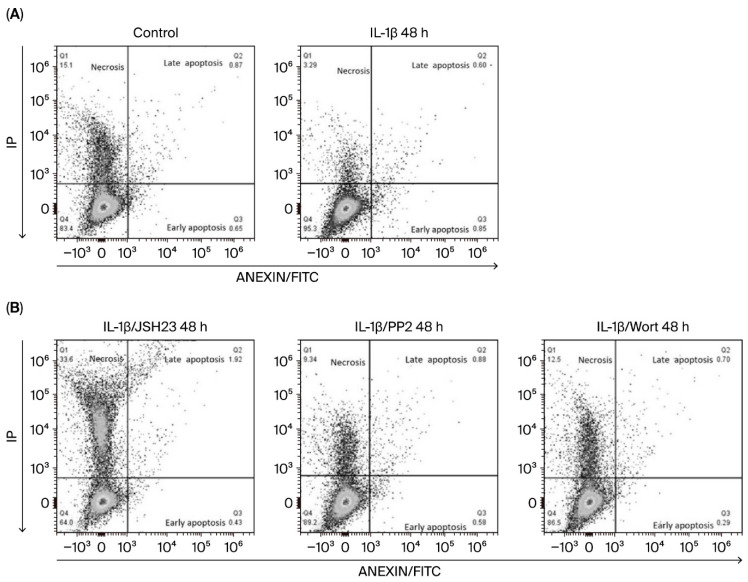
IL-1β inhibits early apoptosis and limits necroptosis in RS4:11 leukaemic lymphoblasts Via PI3K/AKT/Src/NF-κB. (**A**) Representative dot plots showing the analysis of apoptosis in leukaemic lymphoblasts incubated with and without (control) IL-1β at 48 h. The procedure used flow cytometry with an Annexin V probe and labelling with propidium iodide (PI) and fluorescein isothiocyanate (FITC). (**B**) Representative dot plots illustrating apoptosis in leukaemic lymphoblasts incubated for 48 h with IL-1β in the presence of JSH23, PP2 and wortmannin. All results are the mean of three independent experiments, with each experimental condition performed in triplicate.

**Figure 4 biomedicines-14-00041-f004:**
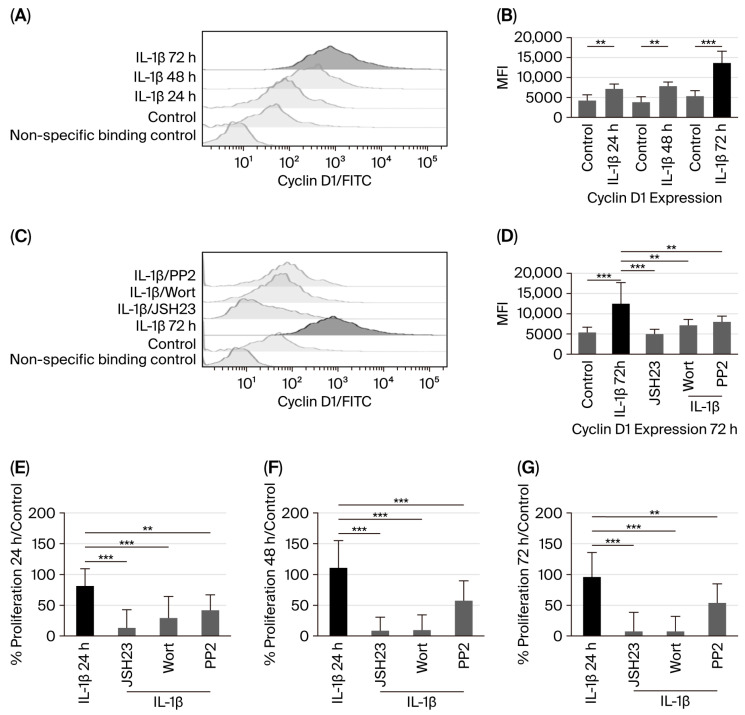
IL-1β induces cyclin D1 overexpression and increases proliferation in RS4:11 leukaemic lymphoblasts via PI3K/AKT/Src/NF-κB. (**A**) Representative histogram of cyclin D1 expression in leukaemic lymphoblasts cultured with IL-1β at 24, 48, and 72 h, measured via flow cytometry. (**B**) MFI analysis of cyclin D1 expression at 24, 48, and 72 h with IL-1β. (**C**) Representative histogram of cyclin D1 expression at 72 h cultured with IL-1β in the presence of inhibitors wortmannin, PP2, and JSH23. (**D**) MFI analysis of cyclin D1 expression at 72 h with IL-1β in the presence of wortmannin, PP2, and JSH23. (**E**) Percentage analysis of leukaemic lymphoblast proliferation after incubation with IL-1β for 24 h and IL-1β in the presence of JSH23, wortmannin, and PP2. (**F**) Percentage analysis of leukaemic lymphoblast proliferation incubated with IL-1β for 48 h and IL-1β plus the same inhibitors. (**G**) Percentage analysis of leukaemic lymphoblast proliferation incubated with IL-1β for 72 h and IL-1β plus the same inhibitors. All results are the means of three independent experiments, each performed in triplicate. *** *p* < 0.01 vs. control, ** *p* < 0.02 vs. control.

**Figure 5 biomedicines-14-00041-f005:**
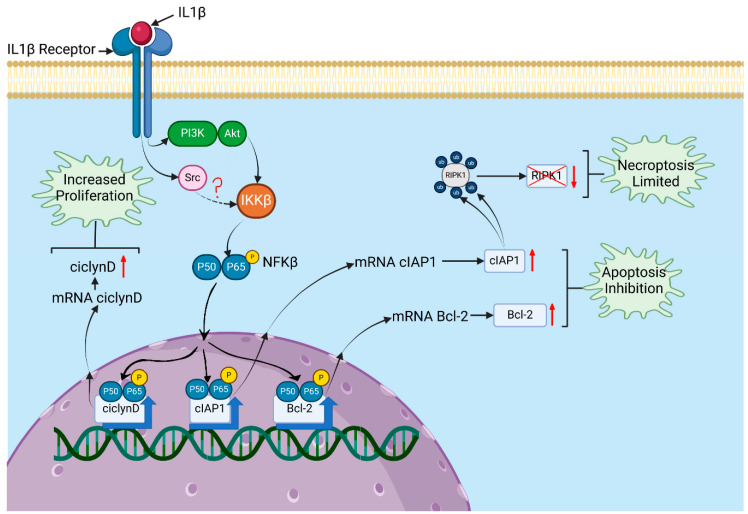
Signalling pathway triggered by IL-1β affecting proliferation and death in RS4:11 leukaemic lymphoblasts. A model is proposed based on the results obtained in this study and previously reported data on the targeting of PI3K/AKT to IKKβ, Src to IKKβ (Src is likely to phosphorylate IKKβ)., and IKKβ to the p65 subunit of NF-κB, leading to its translocation to the nucleus. IL-1β binds to its receptor on leukaemic lymphoblasts, recruiting and activating PI3K/AKT and Src at this receptor. These kinases then activate IKKβ, which phosphorylates the p65 subunit of NF-κB, causing it to translocate to the nucleus. Once in the nucleus, NF-κB induces the transcription of target genes such as cyclin D1, Bcl-2, and cIAP1. Overexpression of cyclin D1 can increase proliferation. Higher levels of Bcl-2 and cIAP1 may be associated with inhibition of apoptosis, and cIAP1 might also help limit necroptosis through the ubiquitination of RIPK1 and its subsequent degradation. Created in BioRender. Gudiño garcía, A. (2025) https://BioRender.com/chl9kru.

**Table 1 biomedicines-14-00041-t001:** IL-1β inhibits apoptosis and limits necroptosis through PI3K/AKT/Src/NF-κB in RS4:11 leukaemic lymphoblasts.

Treatment	Viable Cells (%)	Early Apoptosis (%)	Late Apoptosis (%)	Necrosis (%)
Control	66.11 ± 7.0	5.0 ± 0.08	0.84 ± 0.10	13.54 ± 2.7
IL-1β	89.76 ± 5.0	0.90 ± 0.06	0.91 ± 0.33	5.0 ± 1.9
IL-1β + JSH23	66.47 ± 7.3	0.53 ± 0.12	1.9 ± 6.16	30.5 ± 3.8
IL-1β + PP2	84.90 ± 6.4	0.62 ± 0.10	0.83 ± 0.09	10.53 ± 2.3
IL-1β + Wort	85.50 ± 5.4	0.24 ± 0.05	0.75 ± 0.07	12.7 ± 1.3

## Data Availability

The datasets produced and analysed during this study are available upon reasonable request from the corresponding author.

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
