# Peer review of "IL-1β Controls Proliferation, Apoptosis, and Necroptosis Through the PI3K/AKT/Src/NF-κB Pathway in Leukaemic Lymphoblasts"

_biomedicines, 2025, doi:10.3390/biomedicines14010041_

Round 1
Reviewer 1 Report (Previous Reviewer 2)
Comments and Suggestions for Authors
Thank you for the revisions.
Author Response
Response
All authors are grateful for the reviewer’s comments.
Reviewer 2 Report (New Reviewer)
Comments and Suggestions for Authors
Dr. Victoria-Avila Zl. et al. investigated the effect of IL-1ß for proliferation, apoptosis, and necroptosis of RS4:11 cells which is the cell line derived from a patient with relapsed acute lymphoblastic leukemia (ALL). As the results, the authors showed that IL-1ß can control the cell proliferation or death through the PI3K/AKT/NF-κB signaling pathway in RS4:11 cells although the similar results were observed in another types of neoplasia. The authors showed the interesting results. However, there are some concerns as follows.
1) This is simple comment. The authors should describe that IL-1ß receptors are expressed on RS4:11 cells.
2) The authors should describe that the toxic effects of three inhibitors are not involved in the data showed in Figures. That is, at first, the authors checked about the toxicities of three inhibitors for RS4:11 cells in some way? This point is important in the relation to comment 3) below.
3) Table 1. The authors should discuss about the decrease of early apoptotic cells in which Annexin V expression is used as the indicator even under the inhibition by the inhibitors. In addition, the authors should also discuss about the discrepancy between the data of early apoptosis and necrosis. Is there any possibility that the toxicities of the inhibitors reflect in these data?
4) Discussion section is too long. Should be shorten.
Author Response
Comments and Suggestions for Authors
Dr. Victoria-Avila Zl. et al. investigated the effect of IL-1ß for proliferation, apoptosis, and necroptosis of RS4:11 cells which is the cell line derived from a patient with relapsed acute lymphoblastic leukemia (ALL). As the results, the authors showed that IL-1ß can control the cell proliferation or death through the PI3K/AKT/NF-κB signaling pathway in RS4:11 cells although the similar results were observed in another types of neoplasia. The authors showed the interesting results. However, there are some concerns as follows.
1) This is simple comment. The authors should describe that IL-1ß receptors are expressed on RS4:11 cells.
Response
In consideration of the reviewer’s comment, we have added to the Discussion that RS4:11 lymphoblasts express the IL-1β receptor, which has been highlighted in yellow.
2) The authors should describe that the toxic effects of three inhibitors are not involved in the data showed in Figures. That is, at first, the authors checked about the toxicities of three inhibitors for RS4:11 cells in some way? This point is important in the relation to comment 3) below.
Response
We employed the inhibitors at concentrations previously reported to be non-toxic, and additionally evaluated lymphoblast viability using the trypan blue exclusion assay. This information has been incorporated into the Materials and Methods section and is highlighted in yellow for clarity.
3) Table 1. The authors should discuss about the decrease of early apoptotic cells in which Annexin V expression is used as the indicator even under the inhibition by the inhibitors. In addition, the authors should also discuss about the discrepancy between the data of early apoptosis and necrosis. Is there any possibility that the toxicities of the inhibitors reflect in these data?
Response
With regard to this comment from the reviewer, we would like to note that the decrease in early apoptosis observed in lymphoblasts treated with IL-1β may be attributable to the proteins Bcl-2 and cIAP1 (both of which were also evaluated in this study, and were found to be upregulated in lymphoblasts exposed to IL-1β). Concerning our findings on early apoptosis versus necrosis in lymphoblasts treated solely with IL-1β, we propose that this pattern is likely related to the action of cIAP1. This explanation has been included in the Discussion section of our manuscript and is highlighted in blue.
Regarding the use of the inhibitors, as mentioned in the previous point, we verified that the concentrations employed were non-toxic to the lymphoblasts; therefore, the effects observed on apoptosis and necrosis in lymphoblasts treated with IL-1β in combination with these inhibitors cannot be attributed to potential toxic effects of the inhibitors.
4) Discussion section is too long. Should be shorten.
Response
We agree with the reviewer that the Discussion section is extensive. However, given the number of evaluations and experimental findings presented in this study, we are concerned that substantially shortening this section could compromise the overall clarity and coherence of the work. Nevertheless, we remain open to refining specific parts should the reviewer consider any section unnecessarily detailed.
Reviewer 3 Report (New Reviewer)
Comments and Suggestions for Authors
This manuscript investigates the role of IL-1β in regulating cell death and proliferation in leukaemic lymphoblasts via the PI3K/AKT/NF-κB pathway. To this end, the authors employed the flow cytometry technique to determine the expression levels and phosphorylation status of various proteins, including AKT, Src, p65, Bcl-2, cIAP1, and cyclin D1; the CCK-8 kit to evaluate proliferation; and the Annexin V Kit for apoptosis in the RS4:11 cell line. The authors concluded that the signalling pathway triggered by IL-1β reduces apoptosis and necroptosis while simultaneously increasing the proliferation of leukaemic lymphoblasts.
Overall, the manuscript is well-structured and the topic is presented clearly. I have a few suggestions for the authors.
Abstract:
The authors have not mentioned the type of study. Please specify that this is an in vitro study and include details about the cell line model used.
Additionally, acronyms and abbreviations should be defined when first used in the manuscript, including the abstract.
Keywords:
I would recommend that the authors add the cell line model used to the keywords, as this would make the article more easily accessible to researchers interested in the specific model/experimental approach employed.
Introduction:
- I would recommend that the authors replace some of the existing references or add new ones with more recent 5-year references to better contextualise their work within current research, highlighting the gaps or open questions that the study aims to address.
- Please, define acronyms and abbreviations when first used.
Materials and Methods:
- Protein expression via flow cytometry subsection:
Please add the code of antibodies used to help other researchers to replicate the study with greater accuracy, ensuring transparency and reliability. - Proliferation and apoptosis section:
Please provide a brief description of the proliferation and apoptosis protocol used. - Statistical analysis section:
The authors adopted parametric statistics to compare continuous variables. Please specify the statistic used to test the normal distribution of data. Additionally, please define the p-value threshold considered statistically significant.
Results:
- Figure 1: Figure 1a is the same as 1c. Please replace it with the correct flow cytometric histograms showing AKT phosphorylation in the RS4:11 cell line cultured with IL-1β at 24, 48, and 72 hours.
- Figure 4: Please add the PP2 labeling in Figure 4E. Additionally, the y-axis name in Figure 4G needs to be revised.
Discussion:
- Please move reference number 24 to the correct position (before “considering this evidence, we tested…………….”).
- Although the authors provide a thorough discussion, the analysis does not address the temporal onset of the observed effects. Specifically, it is not addressed why Bcl-2 expression becomes significantly elevated at 48 hours, whereas cIAP1 is upregulated as early as 24 hours, or why the increase in cyclin D1 is most pronounced at 72 hours. These temporal differences merit consideration, as they may provide key insights into the underlying cellular mechanisms and strengthen the overall interpretation of the results.
Author Response
Comments and Suggestions for Authors
This manuscript investigates the role of IL-1β in regulating cell death and proliferation in leukaemic lymphoblasts via the PI3K/AKT/NF-κB pathway. To this end, the authors employed the flow cytometry technique to determine the expression levels and phosphorylation status of various proteins, including AKT, Src, p65, Bcl-2, cIAP1, and cyclin D1; the CCK-8 kit to evaluate proliferation; and the Annexin V Kit for apoptosis in the RS4:11 cell line. The authors concluded that the signalling pathway triggered by IL-1β reduces apoptosis and necroptosis while simultaneously increasing the proliferation of leukaemic lymphoblasts.
Overall, the manuscript is well-structured and the topic is presented clearly. I have a few suggestions for the authors.
Abstract:
The authors have not mentioned the type of study. Please specify that this is an in vitro study and include details about the cell line model used.
Additionally, acronyms and abbreviations should be defined when first used in the manuscript, including the abstract.
Response
We thank the reviewer for this valuable suggestion. We have now explicitly stated that this is an in vitro study conducted using the RS4:11 leukaemic lymphoblast cell line model (highlighted in yellow)
We have now defined all acronyms and abbreviations at their first use throughout the manuscript, including in the abstract, with changes highlighted in yellow for ease of review.
Keywords:
I would recommend that the authors add the cell line model used to the keywords, as this would make the article more easily accessible to researchers interested in the specific model/experimental approach employed.
Response
We have now added the RS4:11 cell line model to the keywords, with the revision highlighted in yellow.
Introduction:
I would recommend that the authors replace some of the existing references or add new ones with more recent 5-year references to better contextualise their work within current research, highlighting the gaps or open questions that the study aims to address.
Please, define acronyms and abbreviations when first used.
Response
We have replaced several existing references with more recent ones to better contextualise our findings within the current research landscape and highlight key gaps addressed by this study, with changes highlighted in yellow.
We have defined all acronyms and abbreviations at their first use throughout the manuscript, with revisions highlighted in yellow
Materials and Methods:
Protein expression via flow cytometry subsection:
Please add the code of antibodies used to help other researchers to replicate the study with greater accuracy, ensuring transparency and reliability.
Response
We have now added catalogue codes for all antibodies used, enabling other researchers to replicate the study, with these details incorporated into the Materials and Methods section and highlighted in yellow.
Proliferation and apoptosis section:
Please provide a brief description of the proliferation and apoptosis protocol used.
Response
We have now provided a brief description of the proliferation and apoptosis protocols used, with these details added to the Materials and Methods section and highlighted in yellow.
Statistical analysis section:
The authors adopted parametric statistics to compare continuous variables. Please specify the statistic used to test the normal distribution of data. Additionally, please define the p-value threshold considered statistically significant.
Response
We employed parametric statistics after confirming the normality of continuous variables using the Shapiro–Wilk test (p > 0.05 in all cases). A p-value < 0.05 was considered statistically significant. These details are now included in the Materials and Methods section and highlighted in yellow.
Results:
Figure 1: Figure 1a is the same as 1c. Please replace it with the correct flow cytometric 72 hours.
Response
We would like to clarify that Figure 1A shows AKT phosphorylation in lymphoblasts cultured with IL-1β at different incubation times. We observed that the highest level of AKT phosphorylation occurred at 24 h; for this reason, Figure 1C presents the inhibition of AKT phosphorylation by the inhibitor wortmannin at the 24 h time point. As the control lymphoblasts displayed very similar basal phosphorylation levels (as reflected in their mean fluorescence intensity) across all culture times, we decided to average these values and present them as a single control.
Figure 4: Please add the PP2 labeling in Figure 4E. Additionally, the y-axis name in Figure 4G needs to be revised.
Response
We have made the requested corrections. The PP2 label has been added to Figure 4E, and the y-axis title in Figure 4G has been revised accordingly.
Discussion:
Please move reference number 24 to the correct position (before “considering this evidence, we tested…………….”).
Response
We have moved reference number 24 to the correct position, and this modification has been highlighted in yellow in the text.
Although the authors provide a thorough discussion, the analysis does not address the temporal onset of the observed effects. Specifically, it is not addressed why Bcl-2 expression becomes significantly elevated at 48 hours, whereas cIAP1 is upregulated as early as 24 hours, or why the increase in cyclin D1 is most pronounced at 72 hours. These temporal differences merit consideration, as they may provide key insights into the underlying cellular mechanisms and strengthen the overall interpretation of the results.
Response
We had originally addressed this observation regarding the time point of highest expression of Bcl-2, cIAP1, and cyclin D in lymphoblasts treated with IL-1β in our manuscript; however, in an effort to shorten the Discussion, we decided to omit it. We have now reinstated this explanation in the Discussion, and it appears highlighted in yellow.
Round 2
Reviewer 3 Report (New Reviewer)
Comments and Suggestions for Authors
Thank you for the revised manuscript. The updated version shows clear improvements. The manuscript is suitable for publication.
This manuscript is a resubmission of an earlier submission. The following is a list of the peer review reports and author responses from that submission.
Round 1
Reviewer 1 Report
Comments and Suggestions for Authors
In the paper by Victoria-Avila et al. the authors describe their experiments in which RS4:11 cells were exposed to L-1β ± signal transduction inhibitors, and afterwards the levels of various intracellular proteins were measured in flow cytometry. In addition, proliferation and apoptosis were measured post-treatment. The goal of the study was “to investigate whether the inflammatory cytokine IL-1β, through kinases such as PI3K/AKT, Src, and NF-κB, affects cell death and proliferation in leukemic lymphoblasts.”
- The novelty of the findings presented in this paper is limited. It has been documented (and cited in references) that IL-1β activates PI3K/AKT, Src, and NF-κB, and that activation of NF-κB in immune cells induces proliferation and resistance to apoptosis. Therefore, this paper may be considered as a confirmation of previous findings in other cells.
- Unfortunately, an important limitation of this paper is that all experiments were performed using just one cell line, and using just one experimental approach, flow cytometry. Moreover, the authors instead of using the name of the cell line, RS4:11, describe these cells “leukemic lymphoblasts” throughout the paper, which suggest using primary cultures.
- The measurements of intracellular proteins using flow cytometry are widely used, specifically when co-expressions of different proteins are studied. However, this method needs suitable controls. According to the Materials and Methods section, the isotype controls were missing in the experiments presented in this paper.
- When the authors present the results of their experiments in the figures, they show just one control, while the graphs show separate controls for each time-frame (e.g. Figure 1a-g; Fig. 2a-f).
- The authors interpret the MFI of the stained cells as the protein level. Moreover, they interpret when the MFI increases 2x that the level of protein is two times higher. This is misinterpretation, specifically in the histograms where obviously two populations, negative and positive overlap.
- Staining of intracellular proteins in flow cytometry is sensitive to many unspecific reactions. Thus, the confirmation that indeed the protein of interest was quantified using western blot should be performed.
- The data presented in Table 1 have not been analysed statistically. Still the authors in the text draw certain unjustified conclusions from these data, for example “we found that this decreased by 1.7-fold when NF-κB was inhibited (0.94% versus 0.53%), by 3.9-fold when PI3K was inhibited (0.94% versus 0.24%)”. These values don’t seem to be significantly different.
- The authors misinterpret necrosis which they detected in flow cytometry as necroptosis.
- The authors write that “it has not yet been established whether IL-1β can affect cyclin D1 expression” which is not true (e.g. https://febs.onlinelibrary.wiley.com/doi/10.1111/j.1432-1033.2004.04385.x).
- In Figure 4 proliferation should be presented as % of untreated control.
- The authors write “that IL-1β in leukemic lymphoblasts has a negative effect only on early apoptosis” and that “IL-1β does not promote late apoptosis”, while these are phenomena which they observe at the end-point of their experiment.
- There are some errors in English which should be corrected.
There are some errors that should be corrected. For example:
Dates are presented as
we not found
IL-1β has even been reported
etc.
Author Response
Answer review 1
Comments and Suggestions for Authors
-In the paper by Victoria-Avila et al. the authors describe their experiments in which RS4:11 cells were exposed to L-1β ± signal transduction inhibitors, and afterwards the levels of various intracellular proteins were measured in flow cytometry. In addition, proliferation and apoptosis were measured post-treatment. The goal of the study was “to investigate whether the inflammatory cytokine IL-1β, through kinases such as PI3K/AKT, Src, and NF-κB, affects cell death and proliferation in leukemic lymphoblasts.”
- The novelty of the findings presented in this paper is limited. It has been documented (and cited in references) that IL-1β activates PI3K/AKT, Src, and NF-κB, and that activation of NF-κB in immune cells induces proliferation and resistance to apoptosis. Therefore, this paper may be considered as a confirmation of previous findings in other cells.
Response
- We agree with the reviewer’s observation that IL-1β activates PI3K/AKT, Src, and NF-κB. However, previous studies have typically reported these findings in various neoplastic cell types, often examining the activation of these proteins independently. In contrast, our study demonstrates, within a single neoplastic cell model, that IL-1β initiates a signalling pathway involving these kinases and the transcription factor, with specific downstream effects. Furthermore, we provide evidence that these kinases collectively contribute to NF-κB activation.
- Unfortunately, an important limitation of this paper is that all experiments were performed using just one cell line, and using just one experimental approach, flow cytometry. Moreover, the authors instead of using the name of the cell line, RS4:11, describe these cells “leukemic lymphoblasts” throughout the paper, which suggest using primary cultures.
Response
- We elected to employ a single cell model as the ATCC specifies that the RS4:11 cell line was established from a patient with Acute Lymphoblastic Leukemia and describes it as an exemplary model for investigating signalling pathways. This information has been duly incorporated into the Materials and Methods section of our manuscript. Given that the ATCC characterises RS4:11 cells as exhibiting the phenotype of leukemic lymphoblasts, we accordingly refer to them as such throughout the text. It is a common practice in the literature to denote cells by their phenotypic characteristics, even when they are derived from established cell lines.
- The measurements of intracellular proteins using flow cytometry are widely used, specifically when co-expressions of different proteins are studied. However, this method needs suitable controls. According to the Materials and Methods section, the isotype controls were missing in the experiments presented in this paper.
Response
- The reviewer’s observation is indeed correct. In response, we have amended the manuscript to include the control used in our assays, which consisted of a non-specific binding control to account for any background signal. This addition has been incorporated into the description of the flow cytometry methodology and is also reflected in the histograms presented in Figures 1, 2, and 4.
4.These revisions clarify the specificity of the staining and ensure the robustness of the data interpretation. When the authors present the results of their experiments in the figures, they show just one control, while the graphs show separate controls for each time-frame (e.g. Figure 1a-g; Fig. 2a-f).
Response
- The reviewer’s observation is accurate. In the representative histograms for each assessment of phosphorylation or protein expression, we have presented a single control representative of the three distinct culture time points. This approach was adopted because the controls across these time points were comparable, and including all would have rendered the figure excessively large and complicated. For the quantitative graphs, we have displayed the mean of the controls from each evaluated culture time point to provide a concise and clear representation.
- The authors interpret the MFI of the stained cells as the protein level. Moreover, they interpret when the MFI increases 2x that the level of protein is two times higher. This is misinterpretation, specifically in the histograms where obviously two populations, negative and positive overlap.
Response
- We appreciate the reviewer’s observation and agree that MFI is commonly used as an indicator of protein expression in flow cytometry, representing the average fluorescence intensity of stained cells relative to controls. However, since flow cytometry data are usually displayed on a logarithmic scale and biological populations may overlap, interpreting MFI as a simple linear measure of protein quantity requires caution. MFI reflects the central tendency of the fluorescence signal (typically the median or geometric mean) of the cell population, rather than a direct molecule-to-molecule quantification of protein.
When populations overlap in flow cytometry, interpreting fold changes in MFI requires careful consideration. Fold change remains a useful relative measure of protein expression differences between samples, but it does not linearly correspond to absolute protein amounts in individual cells. Overlapping populations mean the MFI reflects a combined fluorescence distribution rather than fully distinct groups. In our study, we accounted for this overlap when interpreting fold changes in MFI, using median or geometric mean values within carefully defined gates. This approach aligns with standard flow cytometry practices, acknowledging that fold changes in MFI provide an approximate, not absolute, quantification of expression differences.
6- Staining of intracellular proteins in flow cytometry is sensitive to many unspecific reactions. Thus, the confirmation that indeed the protein of interest was quantified using western blot should be performed.
Response
- With all due respect to the reviewer, we respectfully disagree with the point raised; flow cytometry is a powerful and widely used technique for detecting and quantifying proteins, both on the cell surface and intracellularly, through the use of fluorophore-conjugated antibodies. This method enables the analysis of protein expression at the single-cell level, providing insights into cellular heterogeneity that are not readily obtainable with techniques such as Western blotting, which assess proteins in homogenised cell populations. Furthermore, flow cytometry allows simultaneous detection of multiple proteins and the characterisation of specific cell subpopulations with high sensitivity and speed.
While intracellular protein staining in flow cytometry can be susceptible to nonspecific binding reactions, these issues are minimised through the use of appropriate controls, optimised fixation and permeabilisation protocols, and validated antibodies. Therefore, although confirmation of results by Western blotting is a valid recommendation to verify specificity, it does not detract from the capacity and utility of flow cytometry as a method for analysing protein expression.
In summary, flow cytometry constitutes a valuable complementary tool for studying protein expression, offering single-cell, multiparametric analysis capabilities that Western blotting does not directly provide.
References supporting the afore mentioned statements:
Hedhammar M, Stenvall M, Lönneborg R, Nord O, Sjölin O, Brismar H, Uhlén M, Ottosson J, Hober S. A novel flow cytometry-based method for analysis of expression levels in Escherichia coli, giving information about precipitated and soluble protein.J Biotechnol. 2005 .119(2):133-46. doi: 10.1016/j.jbiotec.2005.03.024
Shi X, Fan W, Mehrpouyan M, Chen Y, D'Cruz LM, Widmann SJ, Tyznik AJ. Cytometry A. Flow cytometry analysis of protein expression using antibody-derived tags followed by CITE-Seq.
2024.105(1):62-73. doi: 10.1002/cyto.a.24792
- The data presented in Table 1 have not been analysed statistically. Still the authors in the text draw certain unjustified conclusions from these data, for example “we found that this decreased by 1.7-fold when NF-κB was inhibited (0.94% versus 0.53%), by 3.9-fold when PI3K was inhibited (0.94% versus 0.24%)”. These values don’t seem to be significantly different.
Response
- We agree with the reviewer and have performed Student’s t-test statistical analyses on the values presented in Table 1. These results are now provided as supplementary material in Tables 2, 3, and 4, where the statistical significance of the values reported in Table 1 can be observed clearly.
- The authors misinterpret necrosis which they detected in flow cytometry as necroptosis.
Response
- We respectfully disagree with the reviewer, as necrosis is, by definition, an unregulated form of cell death. Our data demonstrate that the necrosis induced by IL-1β in lymphoblasts is regulated. Firstly, this is evidenced by the cytokine itself, since IL-1β decreases necrosis compared to the control. Secondly, inhibition of PI3K, Src, and NF-κB increases necrosis. These findings indicate that the “necrosis” observed in our study is controlled by IL-1β and the signalling pathway it activates. Therefore, given that this process is regulated, it should be classified as necroptosis rather than unregulated necrosis.
This distinction aligns with current understanding: necrosis is typically uncontrolled and passive, whereas necroptosis is a programmed, regulated form of necrosis characterized by specific signalling pathways and regulatory mechanisms.
We explain this point in the discussion section of the manuscript pages 18-19.
9- The authors write that “it has not yet been established whether IL-1β can affect cyclin D1 expression” which is not true (e.g. https://febs.onlinelibrary.wiley.com/doi/0).
Response
9- In the manuscript, we included the information from this reference near the end of the Discussion section, page 19.
- In Figure 4 proliferation should be presented as % of untreated control.
Response
- We now present in Figure 4 the proliferation of lymphoblasts expressed as a percentage relative to the proliferation rate of control lymphoblasts.
- The authors write “that IL-1β in leukemic lymphoblasts has a negative effect only on early apoptosis” and that “IL-1β does not promote late apoptosis”, while these are phenomena which they observe at the end-point of their experiment.
Response
- We concur with the reviewer’s assessment and accordingly consider that IL-1β exerts its effects throughout the entire apoptotic process. This interpretation has been incorporated in place of the original concept suggested by the reviewer, page 17. Furthermore, we have revised the manuscript title, substituting “early apoptosis” with “apoptosis.”
- There are some errors in English which should be corrected. Comments on the Quality of English Language. There are some errors that should be corrected. For example:
Dates are presented as: we not found .IL-1β has even been reported etc.
Response
- The manuscript was professionally edited for English language and style by the MDPI author services

Reviewer 2 Report
Comments and Suggestions for Authors
The manuscript presents a focused investigation into the role of IL-1β in regulating apoptosis, necroptosis, and proliferation in leukemic lymphoblasts through the PI3K/AKT/Src/NF-κB pathway. The study contributes to understanding the proinflammatory signaling mechanisms that may sustain leukemic cell survival and proliferation. Several issues should be clarified or expanded before the manuscript can be considered for publication.
Abstract
Please just briefly specify the experimental design including the number of replicates and key assays used such as flow cytometry and CCK-8 and include representative quantitative results.
The conclusion should emphasize how these findings enhance understanding of leukemic pathophysiology or therapeutic implications.
Methods
Provide justification for inhibitor concentrations and selection with supporting literature or preliminary data.
Include the total number of biological replicates in the statistical analysis section and clarify whether ANOVA assumptions of normality and variance were assessed.
Discussion
Discuss the potential clinical or therapeutic implications of targeting IL-1β or this signaling axis in leukemia to enhance the manuscript’s translational value.
Clearly state the strengths and limitations of this study and provide recommendations for future research.
Others
Ensure all figures are of adequate resolution, include clear labeling, and contain scale bars where appropriate.
Maintain consistency in gene and protein nomenclature, including correct formatting of IL-1β, NF-κB, and PI3K/AKT throughout the manuscript.
Author Response
Answer review 2
Comments and Suggestions for Authors
The manuscript presents a focused investigation into the role of IL-1β in regulating apoptosis, necroptosis, and proliferation in leukemic lymphoblasts through the PI3K/AKT/Src/NF-κB pathway. The study contributes to understanding the proinflammatory signaling mechanisms that may sustain leukemic cell survival and proliferation. Several issues should be clarified or expanded before the manuscript can be considered for publication.
Abstract
1.Please just briefly specify the experimental design including the number of replicates and key assays used such as flow cytometry and CCK-8 and include representative quantitative results.
-The conclusion should emphasize how these findings enhance understanding of leukemic pathophysiology or therapeutic implications.
Response
- In response to the reviewer’s comments in the abstract, we now mention the methodology used in our research as well as a conclusion emphasizing how the inflammatory environment can influence the pathophysiology of ALL. Additionally, we state that the proteins involved in the signalling pathway triggered by IL-1β could be considered potential therapeutic targets for this condition.
We apologise to the reviewer, as due to the word limit imposed on the abstract, we decided not to include quantitative results, given that these are numerous.
Methods
- Provide justification for inhibitor concentrations and selection with supporting literature or preliminary data.
Response
- We used PI3K, Src, and NF-κB inhibitors at the concentrations specified in the Materials and Methods section, taking into account references from other studies that employed the same inhibitors, which we have now included in the aforementioned section, page 4.
3, Include the total number of biological replicates in the statistical analysis section and clarify whether ANOVA assumptions of normality and variance were assessed.
Response
- We have provided in the Materials and Methods section the number of experiments conducted, including the replicates for each, and described the statistical analysis applied to each determination carried out in this study, Page 5.
Discussion
- Discuss the potential clinical or therapeutic implications of targeting IL-1β or this signaling axis in leukemia to enhance the manuscript’s translational value.
-Clearly state the strengths and limitations of this study and provide recommendations for future research.
Response
- Within the discussion section of our manuscript, we noted that IL-1β, together with the proteins involved in its downstream signalling pathway, may serve as potential therapeutic targets. In response to the reviewer’s comment, we have expanded on the limitations of our study and suggest that further investigation is warranted in future research, pages 22-23.
Others
- Ensure all figures are of adequate resolution, include clear labeling, and contain scale bars where appropriate.
Response
- We have revised all figures in accordance with the reviewer’s comments.
- Maintain consistency in gene and protein nomenclature, including correct formatting of IL-1β, NF-κB, and PI3K/AKT throughout the manuscript.
Response
- We have addressed the reviewer’s comment by ensuring consistency in gene and protein nomenclature, including the correct formatting of IL-1β, NF-κB, and PI3K/AKT throughout the manuscript.

Round 2
Reviewer 1 Report
Comments and Suggestions for Authors
In their answers to the Reviewer, the Authors of the paper mostly argument why the Reviewer was wrong. They did not address the most important issues. Therefore, the paper in this form is not suitable for publication.
Author Response
In their answers to the Reviewer, the Authors of the paper mostly argument why the Reviewer was wrong. They did not address the most important issues. Therefore, the paper in this form is not suitable for publication.
Response:
We extend our apologies for instances in which we may not have adequately explained or discussed our perspective regarding certain reviewer comments. As we have incorporated the changes suggested by the reviewers, we now address those points for which our rationale may not have been sufficiently clear in relation to the observations made:
- Unfortunately, an important limitation of this paper is that all experiments were performed using just one cell line, and using just one experimental approach, flow cytometry. Moreover, the authors instead of using the name of the cell line, RS4:11, describe these cells “leukemic lymphoblasts” throughout the paper, which suggest using primary cultures.
Response
We considered employing a cellular model that would represent the majority of cases presenting acute lymphoblastic leukaemia; the RS4:11 cell line meets these requirements (as outlined in the previous response). However, we decided against using an alternative cell line owing to the potential variability in the response to IL-1β, which might have resulted in inconclusive findings. Using a single-cell-line model such as RS4:11 offers several advantages, including the provision of a consistent and reproducible system characterised by uniform genetic and phenotypic features. It also simplifies interpretation by reducing biological variability and avoids confounding effects arising from differential pathway responses between cell lines. This approach enables efficient use of time and resources, facilitates direct comparison with existing literature, and ensures more reliable conclusions regarding IL-1β signalling in acute lymphoblastic leukaemia. Nevertheless, it may limit broader applicability and overlook cell-line-specific responses. Overall, employing a single well-characterised model such as RS4:11 provides an appropriate balance between rigour and practicality for mechanistic studies.
Regarding the rationale for referring to RS4:11 cells as leukaemic lymphoblasts, this was addressed in our initial response. To avoid any ambiguity on this point, we have now clarified in the results subheadings, in each figure legend, and at the beginning of the discussion that we are referring to leukaemic lymphoblasts derived from the RS4:11 cell line.
On the other hand, regarding the use of a single method for determining protein phosphorylation and expression—flow cytometry in our case—we agree with the reviewer that these analyses can also be performed using immunoblotting. In our view, neither technique invalidates the other (the reasons for this have already been explained in the previous letter).In our particular case, we employed this technique because flow cytometry offers several technical advantages, including single-cell resolution that enables the detection of heterogeneous signalling responses; rapid, high-throughput analysis that improves reproducibility; reduced sample requirements, which is advantageous when material is limited; and lower susceptibility to artefacts, as immediate fixation minimises dephosphorylation during sample processing.
- The authors interpret the MFI of the stained cells as the protein level. Moreover, they interpret when the MFI increases 2x that the level of protein is two times higher. This is misinterpretation, specifically in the histograms where obviously two populations, negative and positive overlap.
Response
The MFI obtained by flow cytometry serves as an indicative measure of protein expression, since the fluorescent signal is proportional to the amount of antibody bound to the target protein. Therefore, higher protein abundance leads to increased fluorescence intensity. MFI reflects the average fluorescence per cell within a defined population, allowing reliable semiquantitative comparisons between experimental conditions. This method is widely accepted for assessing both total and phosphorylated proteins, as flow cytometry provides single-cell analysis with enhanced sensitivity to detect subtle changes in expression or activation. MFI remains a reliable metric even when two populations partially overlap in flow cytometry, as long as the population of interest is delineated using a robust gating strategy. Because flow cytometry analyses cells individually, the MFI calculation includes only events within the defined gate. Consequently, even if populations overlap visually in histograms or dot plots, the MFI is valid when gating effectively minimises cross-contamination.
- The data presented in Table 1 have not been analysed statistically. Still the authors in the text draw certain unjustified conclusions from these data, for example “we found that this decreased by 1.7-fold when NF-κB was inhibited (0.94% versus 0.53%), by 3.9-fold when PI3K was inhibited (0.94% versus 0.24%)”. These values don’t seem to be significantly different.
Response
To further clarify this point, we now indicate in the results referring to Table 1 that the significant differences between the values being compared are presented in the Supplementary Materials.
Finally, all authors of the present work are grateful for the reviewer’s comments, most of which have led to modifications that have improved and enriched the manuscript. We remain attentive and open to any further observations that may arise from this revised version of our work.
